# Robust Immune Response and Protection against Lethal Pneumococcal Challenge with a Recombinant BCG-PspA-PdT Prime/Boost Scheme Administered to Neonatal Mice

**DOI:** 10.3390/vaccines12020122

**Published:** 2024-01-25

**Authors:** Monalisa Martins Trentini, Dunia Rodriguez, Alex Issamu Kanno, Cibelly Goulart, Michelle Darrieux, Luciana Cezar de Cerqueira Leite

**Affiliations:** 1Laboratório de Desenvolvimento de Vacinas, Instituto Butantan, São Paulo 05503-900, Brazil; 2Laboratório de Microbiologia Molecular e Clínica, Universidade São Francisco, Bragança Paulista 12916-900, Brazil; michelle.bertoncini@usf.edu.br

**Keywords:** recombinant BCG, neonatal mice, streptococcus pneumoniae, prime/boost strategy

## Abstract

Pneumococcal diseases are an important public health problem, with high mortality rates in young children. Although conjugated pneumococcal vaccines offer high protection against invasive pneumococcal diseases, this is restricted to vaccine serotypes, leading to serotype replacement. Furthermore, the current vaccines do not protect neonates. Therefore, several protein-based pneumococcal vaccines have been studied over the last few decades. Our group established a recombinant BCG expressing rPspA-PdT as a prime/rPspA-PdT boost strategy, which protected adult mice against lethal intranasal pneumococcal challenge. Here, we immunized groups of neonate C57/Bl6 mice (6–10) (at 5 days) with rBCG PspA-PdT and a boost with rPspA-PdT (at 12 days). Controls were saline or each antigen alone. The prime/boost strategy promoted an IgG1 to IgG2c isotype shift compared to protein alone. Furthermore, there was an increase in specific memory cells (T and B lymphocytes) and higher cytokine production (IFN-γ, IL-17, TNF-α, IL-10, and IL-6). Immunization with rBCG PspA-PdT/rPspA-PdT showed 100% protection against pulmonary challenge with the WU2 pneumococcal strain; two doses of rPspA-PdT showed non-significant protection in the neonates. These results demonstrate that a prime/boost strategy using rBCG PspA-PdT/rPspA-PdT is effective in protecting neonates against lethal pneumococcal infection via the induction of strong antibody and cytokine responses.

## 1. Introduction

Pneumococcal infections remain a significant global public health problem worldwide, responsible for more than 740,000 deaths among children under 5 years of age in developing countries [1,2]. Pneumococcal conjugate vaccines are licensed for use in children, providing protection against up to 13 prevalent pneumococcal serotypes (PCV7, PCV10, and PCV13) [3,4,5,6]. While PCVs have a very low failure rate in children from industrialized countries with effective national immunization programs, the limited vaccination coverage in many parts of the world, replacement by non-vaccine serotypes, and increasing resistance to antibiotics greatly impact the efficacy of these formulations in low- and middle-income countries (LMIC), reinforcing the need to develop alternative vaccines [7,8]. The highly complex production process for PCVs leads to high production costs and limits their production in LMIC. Therefore, an ideal pneumococcal vaccine should induce broad cross reactivity (serotype-independent) and have low production costs [9,10,11]. 

Many research investigations have shown the promise of pneumococcal proteins as possible alternatives for developing serotype-independent vaccines against pneumococcal disease [12,13,14,15]. Pneumococcal surface protein A (PspA) has been extensively investigated and considered a promising vaccine candidate, displaying high immunogenicity and protective efficacy in different infection models [16,17,18,19]. PspA is an exposed virulence component that generates antibodies with the capacity to increase complement activation, C3 deposition, and killing via opsonophagocytosis [20,21,22,23]. PspA has also been shown to protect the bacterium from antimicrobial peptides derived from lactoferrin [24]. PspA has been shown to display some degree of variability, classified into three families comprising five clades [13,25,26]. We previously determined that most isolated strains in Brazil are distributed between families 1 (comprising clades 1 and 2) and 2 (clades 3 and 4) [27]; this is also true for other countries. Furthermore, we determined that the induction of serocross reactivity was more efficient within families [16]. Therefore, we selected PspA molecules inducing broad-range cross reactivity within family 1 [28] and family 2 [23]. 

Pneumolysin (Ply) is a cholesterol-dependent cytolysin that activates complement, induces apoptosis in different host cells, and promotes inflammatory responses and lung injury [29,30,31,32,33,34]. It is the main toxin produced by pneumococci and an important virulence factor. Ply is a TLR4 ligand that not only induces innate immune responses but also induces the TLR4-independent activation of inflammasomes [35,36], contributing to host protection. While Ply is naturally toxic, modified molecules, which are denominated pneumolysoids, display low toxicity while retaining the immunogenic properties of the original protein. PdT (a genetically detoxified form of Ply) is a well-characterized pneumolysoid that has been demonstrated to stimulate cells via Toll-like receptor 4 (TLR4) in a similar fashion to native pneumolysin [35,37].

The immunization of infants with PCVs includes 3–4 doses starting at 2 months, which renders neonates susceptible to infection in the first months of life, where mortality is higher. Partial protection is attained by reducing transmission from vaccinated older children and adults. Therefore, there are efforts to develop vaccines that would induce protection in newborns [38,39]; however, there are very few. 

BCG is one of the only vaccines that induces protection in newborns. Our previous results showed that recombinant BCG (rBCG) expressing *Bordetella pertussis* antigens induced protection against challenge in neonate mice, contrary to the conventional vaccine [40]. We previously constructed a fusion protein comprising a PspA with PdT. This hybrid was able to induce antibodies that can adhere to the surface of pneumococci of different serotypes, promoting protection against lethal pneumococcal challenge and pneumonia [41]. Subsequently, we developed a rBCG strain expressing the rPspA-PdT fusion protein (rBCG PspA-PdT). When administrated in a prime/boost strategy, the rBCG PspA-PdT/rPspA-PdT vaccine induced elevated levels of antibody, with an IgG1/IgG2c antibody isotype shift, and effectively protected mice against a lethal pneumococcal challenge [42]. Furthermore, the immunized mice induced a significant increase in the binding of IgG2c and an improved deposition of complement on the pneumococcal surface in BALF samples, promoting an early clearance of pneumococci [43].

In this context, we investigated the immunogenicity and protective effect of the prime/boost strategy with rBCG PspA-PdT in neonatal mice. 

## 2. Materials and Methods

### 2.1. Pneumococcal Strain and rBCG Strain

*Streptococcus pneumoniae* strain WU2 (serotype 3; PspA clade 2; family 1), generously provided by Dr. David Briles (University of Alabama at Birmingham, Birmingham, AL, USA), were cultivated using the methods previously described [41] and stored at −80 °C. Serotype 3 strains are a major cause of severe clinical manifestations of pneumococcal diseases in humans, exhibiting a wide geographical distribution [44,45]. Furthermore, the WU2 strain is also highly lethal in mice, providing a good model of disease [46]. 

We used an rBCG strain expressing a PspA clade 2, family 1 (previously selected for broad cross reactivity within family 1 and different than the WU2 PspA clade 2), in fusion with PdT, previously constructed in our lab [42]. Briefly, BCG Pasteur was transformed by electroporation with the pMIP12-pspA2-pdT mycobacterial expression vector and vaccines prepared [42].

### 2.2. Neonatal Mice Immunization

The Ethics Committee at Instituto Butantan, São Paulo, Brazil (CEUAIB) approved all animal experiments (Permit Number 9343010422). Groups of litters comprising 6–10 neonate C57BL/6 were immunized intraperitoneally with 50 μL containing 1 × 10^5^ CFU (1/10th adult mouse dose) of rBCG PspA-PdT or WT-BCG on the 5th day after birth. A booster containing 5 μg of rPspA-PdT (and 50 μg of Al(OH)_3_ (Alum) in saline, as adjuvant) was administered 7 days following the priming dose (on the 12th day after birth). Control groups received either saline (negative control) or two doses containing 5 μg of rPspA-PdT and 50 μg of Alum (positive control) (on days 5 and 12 after birth) (also i.p.). 

### 2.3. Lethal Pneumococcal Challenge

All mice groups were under anesthetized 9 days after the protein boost (on day 21st after birth) before receiving 10^6^ CFU of the pneumococcal strain WU2 in 30 µL of saline using intranasal aspiration. Survival was observed for a total of 15 days. Mice that were in a moribund state or exhibited any indications of illness were euthanized. 

### 2.4. Antibodies Measurement

Nine days after the protein boost (on day 21st after birth), sera were obtained from mice via retro-orbital collection, and anti-rPspA-PdT antibodies were evaluated using ELISA. The ELISA assay was conducted as previously described [26]. The absorbance was analyzed using an ELISA reader (Multiscan EX—Uniscience, Helsinki, Finland) at OD_450nm_. The standard curve for calculating the antibody concentration in each sample was established using purified mouse IgG, IgG1, or IgG2a (Southern Biotechnology, Birmingham, AL, USA).

### 2.5. Antibody Binding to Pneumococcal Surface

The evaluation of the binding ability of antibodies from the sera or bronchoalveolar lavage fluid (BALF) of immunized mice to PspA exposed on the pneumococcal surface was assessed as previously described [26]. The WU2 strain was cultured using undiluted sera or BALF samples obtained from immunized mice. Subsequently, the strain was incubated using FITC-conjugated IgG antibody (MP Biomedical) at a dilution of 1:500 in PBS for 30 min. The bacteria were rinsed and placed back into a solution containing 1% paraformaldehyde (PFA). Unstained pneumococci were utilized as a negative control. The samples were then analyzed via flow cytometry using a FACS Canto II (BD, Bioscience, San Jose, CA, USA). 

### 2.6. Spleen and Lungs Cell Culture

Nine days after the protein boost (on day 21 after birth), the animals were euthanized, and the spleens and lungs were collected aseptically. The cell preparations were conducted as previously described [26]. The cell suspensions were adjusted to 1 × 10^6^ cells/mL and incubated with anti-CD28 (1 µg/mL, clone: CD82.2, BD Pharmingen™), anti-CD3 (1 µg/mL, clone: OKT3, BD Pharmingen™), and rPspA-PdT (5 µg/mL) for 48 h at 37 °C and 5% CO_2_. The supernatant sample was collected, and the concentrations of cytokines (IFN-γ, IL-17, TNF-α, IL-10, and IL-6) were evaluated via Cytometric Bead Array (BD Pharmingen™) using the Mouse Th1/Th2/Th17 Cytokine Kit.

Additionally, to investigate the phenotype of memory B and T cells in the lungs and spleen, the remaining cells were labeled with specific antibodies: (i) Memory B cells: anti-B220-FITC antibody (clone: RA3-6B2, BD Pharmingen™), anti-CD19-BV421 antibody (clone: 1D3, BD Horizon™), and anti-CD27-PerCP.Cy5.5 antibody (clone: LG.3A10, BD Pharmingen™); (ii) Memory T cells: anti-CD4-APC.Cy7 antibody (clone: GK1.5, BD Pharmingen™), anti-CD8-PE.Cy7 antibody (clone:53-6.7, BD Pharmingen™), anti-CD62L-FITC antibody (clone: MEL-14, BD Pharmingen™), and anti-CD44-APC antibody (clone: IM7, BD Pharmingen™) for 30 min. Also, the cells were rinsed and resuspended in 1% PFA. Data were acquired using a FACSCanto II flow cytometer.

### 2.7. Bronchoalveolar Lavage Fluid (BALF) Collection and Cytokine Analysis

For BALF collection, a group of immunized mice was euthanized 3 days after the challenge with the WU2 strain (24 days after birth). The trachea was surgically opened, and a catheter was inserted for cannulation. The lungs were washed twice using cold PBS. The samples were stored on ice and centrifuged at 280× *g* for 10 min, and the supernatants sample were used to measure antibody binding and cytokine production. 

The measurement of cytokines (IFN-γ, IL-17, TNF-α, IL-10, and IL-6) in the BALF supernatant samples was performed using Cytometric Bead Array (BD Pharmingen™) using the Mouse Th1/Th2/Th17 Cytokine Kit, according to the manufacturer’s recommendations.

### 2.8. Statistical Analysis

The collected samples were individually evaluated and presented as means ± SD. Statistical analyses were conducted using a one-way ANOVA and Bonferroni’s Multiple Comparison Test between the groups. Survival rates in each group were analyzed using the Kaplan–Meier test; *p* < 0.05 indicated statistical significance.

## 3. Results

### 3.1. Prime/Boost Immunization Using rBCG PspA-PdT and rPspA-PdT Promotes IgG1/IgG2c Antibody Isotype Class Shift and Memory B Cells in Neonatal Mouse Model

To evaluate the anti-PspA-PdT antibody production induced by the prime/boost scheme, neonatal mice were vaccinated with rBCG PspA-PdT on day 5 and received a booster with rPspA-PdT (and Al(OH)_3_ as adjuvant) at day 12). Sera from the immunized mice were evaluated to produce anti-rPspA-PdT antibodies. The groups of mice that received the rBCG PspA-PdT/rPspA-PdT (prime/boost scheme) showed increased total IgG production, comparable to those that received two doses of rPspA-PdT (Figure 1A). The other immunization schemes did not induce significant antibody levels. Mice immunized with rPspA-PdT showed a predominant production of the IgG1 isotype antibodies. Interestingly, mice immunized with rBCG PspA-PdT/rPspA-PdT promoted an IgG class shifting from IgG1 to IgG2c isotype, as seen in adult mice (Figure 1B).

As for the capacity of the antisera to adhere to PspA that is visible on the pneumococcal surface, it was observed that antisera obtained from mice immunized with rBCG PspA-PdT/rPspA-PdT prime/boost exhibited increased IgG binding to pneumococci (strain WU2) (Figure 1C). Surprisingly, the antisera obtained from animals that were administered only rBCG PspA-PdT also exhibited increased binding, although they presented much lower anti-rPspA-PdT antibody levels. However, sera obtained from animals that received two doses of rPspA-PdT (two doses) demonstrated lower binding on the pneumococcal surface, even though it induced high levels of IgG in the ELISA assay (Figure 1A).

Since significant levels of antibodies (IgG and IgG2c) were observed in mice immunized with the prime/boost scheme, the induction of B cells and memory B cells were evaluated in the spleen and lungs of immunized animals, 9 days after the rPspA-PdT booster (gating strategy in Appendix A). All groups of immunized mice displayed an induction of B cells in the spleen (Figure 2A) and lungs (Figure 2B) in comparison to the control group. Vaccination with rBCG PspA-PdT/rPspA-PdT was particularly effective and induced higher levels of B cells. Analysis of the lungs showed significantly higher B cell counts in the prime/boost scheme than in the group receiving two doses of protein. B cells showed a higher percentage in the spleens as compared to the lungs. This is expected since the spleen is a hemopoietic organ with the function of maturation and storage of immune cells. 

When analyzing the induction of memory B cells in the spleens and lungs, a similar and more pronounced phenomenon was observed (Figure 2C,D); the rBCG PspA-PdT/PspA-PdT-immunized mice displayed a significant increase in memory B cells compared with all other control groups. Furthermore, the group immunized with only rBCG PspA-PdT also displayed increased memory B cells compared to the controls. 

### 3.2. Prime/Boost Immunization Using rBCG PspA-PdT and rPspA-PdT Induces Inflammatory Cytokine and Memory T Cells in Neonatal Mouse

Cytokine secretion was assessed in the supernatant of spleen and lung cells cultured with rPspA-PdT. Mice receiving rBCG PspA-PdT/rPspA-PdT induced significantly higher levels of IL-6 (Figure 3A), IL-17 (Figure 3D), and IL-10 (Figure 3E) in the lungs compared to the other immunized mice. The levels of TNF-α were also elevated compared to most groups (Figure 3B). Additionally, the levels of IFN-γ were either higher or comparable in the prime (rBCG PspA-PdT) and rBCG PspA-PdT/rPspA-PdT (prime/boost) groups (Figure 3C). When analyzing the data in a radar chart, it becomes clear that the prime/boost rBCG PspA-PdT/rPspA-PdT group shows increased overall cellular immune responses (Figure 3F).

In contrast, the unstimulated cells obtained from mice that were immunized with prime/boost strategy (rBCG PspA-PdT/rPspA-PdT) exhibited elevated levels of IFN-γ, IL-17, TNF-α, IL-10, and IL-6 in the lungs (Appendix A) compared to the stimulated groups. Furthermore, the spleen cells obtained from groups that received rBCG PspA-PdT/rPspA-PdT, when stimulated with rPspA-PdT, showed significantly higher levels of IL-6, TNF-α, IL-17, and IL-10 (Appendix A) when compared to the other immunized groups. The animals immunized only with the prime rBCG PspA-PdT also showed higher levels of IFN-γ, TNF-α, IL-10, and IL-6 (Appendix A).

However, vaccination with prime/boost rBCG PspA-PdT/rPspA-PdT showed high levels of inflammatory cytokines; we evaluated the induction of subsets of Memory T cells in the spleen and lungs of immunized mice. Those mice that received the rBCG PspA-PdT or rBCG PspA-PdT/rPspA-PdT had a greater increase in the production of Central Memory T cells (TCM) and Effector Memory T cells (TEM) of both CD4 and CD8 phenotypes (Figure 4A–D). 

### 3.3. rBCG PspA-PdT/rPspA-PdT Protects Neonatal Mice against Lethal Pneumococcal Challenge

The efficacy of vaccination with rBCG PspA-PdT/rPspA-PdT in neonate mice was assessed using a lethal pulmonary challenge. Seventy-two hours after the challenge, the BALF was recovered and analyzed from cytokine production. All groups of immunized mice had decreased levels of IL-6, IFN-γ, and IL-17 in the lung cells after infection (Figure 5A,C,D). Interestingly, only mice that received rBCG PspA-PdT (prime or prime/boost) displayed elevated levels of TNF-α and IL-10 (Figure 5B,E). 

At this time point, BALF samples were collected to evaluate the capacity of the bind to the pneumococcal surface. BALF samples obtained from animals that received rBCG PspA-PdT/rPspA-PdT displayed significantly greater levels of total IgG binding to pneumococci (~29% of positive cells). This was followed by rBCG PspA-PdT (~19% of positive cells) (Figure 6A,B). Mouse survival was monitored for 15 days, showing that only rBCG PspA-PdT/rPspA-PdT provides 100% protection against fatal challenges (Figure 6C). Interestingly, although the prime/boost with protein alone showed comparable levels of anti-PspA-PdT antibodies as those primed with rBCG, the protection observed was much lower. Furthermore, mice that received rBCG PspA-PdT (prime dose) or WT-BCG/rPspA-PdT presented partial protection in this model of neonatal mice. 

## 4. Discussion

Despite the successful implementation of conjugate pneumococcal vaccines in many countries, there remain important limitations, including serotype replacement and geographic variations in serotype distribution, which compromise the long-term success of such formulations. Proteins are considered the most effective means of attaining broad cross-reacting immunity. However, proteins have not shown sufficient immunogenicity in humans, even using different adjuvant formulations. Therefore, new presentation systems should be investigated. 

BCG has been used in newborn vaccination against tuberculosis for over a hundred years, displaying high safety; it is one of the few vaccines that can induce protective immunity in neonate children. BCG has shown the induction of long-lasting immunity in humans, and the recombinant BCG strategy could show interesting results. Its potent adjuvanticity reinforces its use as a platform to deliver heterologous proteins from different pathogens. Previously, we showed in a prime/boost strategy that the rBCG PspA-PdT/rPspA-PdT immunization scheme induced high levels of antibody, with an IgG1/IgG2c antibody isotype shift [42] and high protection in adult mice. The present study investigated the use of this prime/boost strategy as a vaccine targeting newborns. The prime/boost strategy in the neonate mouse model induced similar antibody levels in comparison with the recombinant protein alone (Figure 1A). The presence of rBCG in the priming dose promoted an antibody shift to IgG2c (Figure 1B). This isotype change has been previously observed in an adult model [42]. In both adults and neonate mice, immunization with rPspA (in Alum) primarily showed an IgG1 isotype profile, which is protective in adults but not neonates [42].

Furthermore, in the adult model, IgG2a/c and IgG2b antibodies present in the serum and the BALF from mice immunized with the prime/boost strategy also showed increased binding to the pneumococcal surface, indicating that the vaccine antibodies can recognize the native proteins expressed by the bacterium [42,43,47,48]. In the neonatal model, we observed comparable results, showing that the prime/boost strategy induced high levels of IgG2c antibodies, and the serum and BALF were able to bind to the pneumococcal surface, promoting an early clearance of pneumococci (Figure 1C,D; Figure 6A,B). Unfortunately, we did not have a sufficient amount of BALF sample to measure IgG isotypes. 

Most interesting is that although the immunization with rBCG-PspA-PdT only induced lower levels of antibodies to the fusion protein, these antibodies showed a more balanced proportion of IgG1/IgG2c and high binding activity to the pneumococci surface. These results may partly explain the higher protection induced by rBCG-PspA-PdT.

Regarding cytokine production, vaccination with the prime/boost scheme in the neonatal model induced an increase in the secretion of IL-6, IL-17, TNF-α, and IFN-γ in lung cells (Figure 3). Interestingly, these cytokines have been described as protective against pneumonia. IL-17 is considered important in the recruitment of neutrophils to the site of the infection, correlating with the highest protection level, mainly in a mouse model of pneumococcal colonization [49,50,51]. The secretion of IFN-γ and TNF-α via lung cells has been correlated with protection against pneumonia [52,53,54]. In oral administration of non-recombinant *Lactobacillus casei* [55], the balance of the induction of cytokines TNF-α and IL-10 was related to the protection against lung injuries caused by *S. pneumoniae* infection due to the rapid increase in the infiltration of neutrophils. 

Another interesting result is the higher protection observed via immunization with WT-BCG/rPspA-PdT. This has been previously observed in adult mice [43]. This effect has been attributed to the recently described innate immune memory (or trained immunity) properties of BCG, where increased protection against heterologous pathogens has been characterized. However, this protection is not expected to be long-lasting. In fact, this group should lower the production of memory B cells (Figure 2). 

Besides the generation of specific antibodies in serum and cytokines in the lung, priming with rBCG PspA-PdT also resulted in enhanced production of memory B- and T-cells, including TCM and TEM cell populations, in the spleen and lungs (Figure 4). Previous studies have shown that regardless of parenteral or intranasal administration, the memory B and T cell populations residing in the lungs can mediate protection by enhancing the production of pneumococcus-reactive antibodies in the lungs and generating neutrophil-dependent protection against pneumococcal nasal challenge [56,57,58,59]

A significant reduction in cytokine production was observed in the lungs of prime/boost-immunized mice following intranasal challenge (Figure 5). This correlated with the 100% survival after the challenge (Figure 6), suggesting that a more controlled inflammation is important for protection against pneumococcal sepsis. Similar results have been described in previous studies investigating PspA-based vaccines [54]. The subcutaneous immunization of adult mice led to a reduction in IL-6 and IFN-γ and the induction of TNF-α post-challenge in the lungs [54]. Our group has previously shown that the prime/boost strategy provided 100% protection in adult mice after pneumococcal lethal challenge [42]. 

One of the limitations of this study is the reduced number of studies with neonates in the literature, especially in the BALF, which provided few options for proper sampling design. We adopted alternative designs, more appropriate for our objectives, which, unfortunately, reduces the possibility of comparison with previous data. Another important factor was the fact that obtaining a reasonable amount of neonate mice at the same time requires very careful coordination for their birth to be able to obtain several litters at the same time. Furthermore, the amount of blood and BALF obtained is much more limited than in adult mice, restricting the number of samples available for analysis. Another important limitation is the coverage provided by the PspA component of the antigen. Our studies are based on a PspA fragment from family 1, previously demonstrated to induce broad protection within family 1 strains [28]. Although PdT is highly conserved in all pneumococcal strains, it is not expected to provide high protection, working mostly as an adjuvant. In order to attain wide coverage, our strategy would be to include another rBCG strain expressing a family 2 PspA. We previously selected a family 2 PspA inducing broad cross-reactive protection [23] and a fusion protein of this family 2 PspA with PotD, induced protection against invasive challenge and colonization with strains containing PspA family 1 and 2 [60]. Our plan is to investigate the prime/boost strategy with a mixture of rBCG-PspA1-PdT and rBCG-PspA2-PotD, which has the potential to induce protection against invasive challenge and colonization in neonate mice.

## 5. Conclusions

The results of the present study demonstrate that rBCG PspA-PdT is an effective platform for the delivery of pneumococcal antigens, and the prime/boost immunization promotes strong antibody and cell-mediated responses, generating memory T and B cells and conferring protection against invasive infection in neonate mice. 

BCG vaccine has been employed for more than a century with the goal of protecting newborns against tuberculosis, demonstrating the induction of long-lasting immunity and a commendable safety profile. Recombinant BCG vaccines are expected to carry the main properties of BCG and therefore have the potential to induce long-term protection. Our results reinforce the potential of the rBCG-expressing pneumococcal proteins in a prime/boost strategy to be further investigated in humans.

## Figures and Tables

**Figure 1 vaccines-12-00122-f001:**
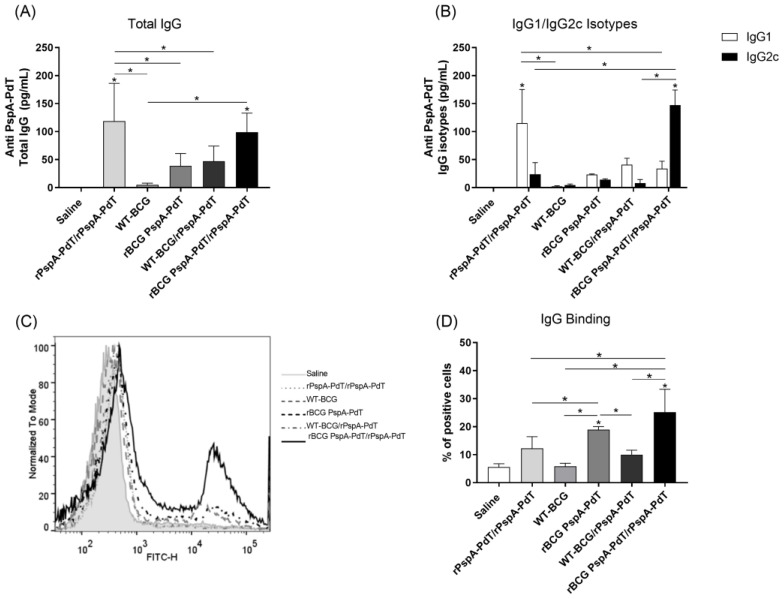
The administration of rBCG PspA-PdT/PspA-PdT to neonate mice induces IgG1/IgG2c antibody isotype shift and IgG binding on the pneumococcal surface. Neonate mice were vaccinated with WT-BCG, rBCG PspA-PdT, or rPspA-PdT. After 7 days, mice received a booster dose with the rPspA-PdT. The rPspA-PdT group was administered two doses of the recombinant protein. The ELISA assay was used to assess the antibodies against rPspA-rPdT 9 days after the booster. (**A**) Total IgG against rPspA-PdT; (**B**) IgG1 and IgG2c against rPspA-PdT. (**C**,**D**) Pneumococcal WU2 strain was cultured using heat-inactivated sera obtained from immunized mice followed by incubation with FITC-conjugated antibody with mouse IgG. The percentage of positive cells expressing FITC-IgG was determined using FACS Canto II (BD Biosciences). The negative control consisted of sera obtained from mice that received only saline. * *p* values < 0.05 were considered statistically significant.

**Figure 2 vaccines-12-00122-f002:**
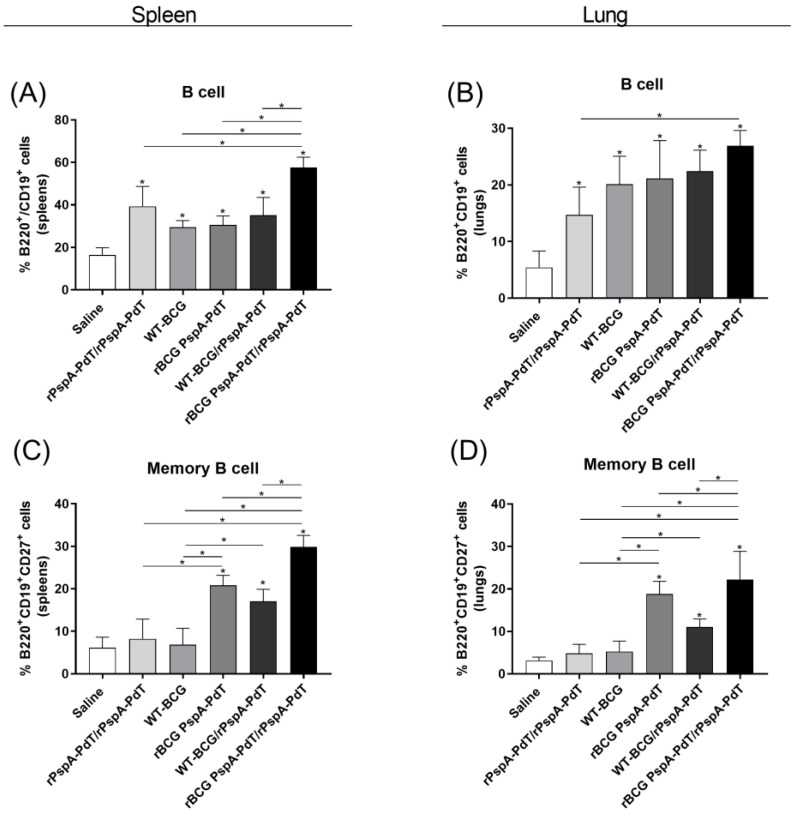
The administration of rBCG PspA-PdT/PspA-PdT elicits the development of Memory B cells in the spleens and lungs of neonate mice. Neonatal mice were immunized with rBCG PspA-PdT, WT-BCG, or rPspA-PdT. After 7 days, mice received a booster dose with the rPspA-PdT. After 9 days of the booster, B cells and Memory B cells in the spleens and lungs were evaluated. B cells were identified as B220^+^CD19^+^ in the spleens (**A**) and the lungs (**B**). Memory B cells were identified as B220^+^CD19^+^CD27^+^ in the spleens (**C**) and the lungs (**D**). For analyses, Flow Cytometer FACS Canto II was used. * *p* values < 0.05 were considered statistically significant. Results are represented by the means ± SD.

**Figure 3 vaccines-12-00122-f003:**
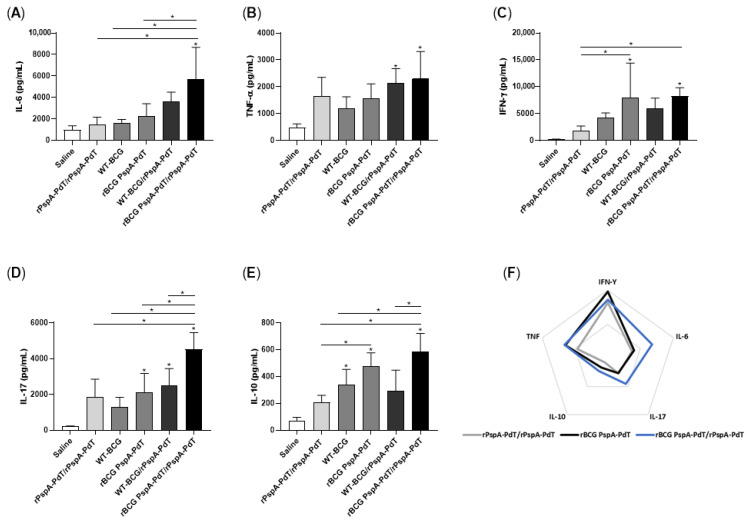
Vaccination with rBCG PspA-PdT/PspA-PdT produces inflammatory cytokines in the lungs of neonate mice. Lung cells obtained from immunized mice 9 days post-administration of the booster dose were cultured for 48 h with rPspA-PdT. Cytokine levels were quantified by mouse Th1/Th2/Th17 cytometric bead array (CBA). Cytokine levels of IL-6 (**A**), TNF-α (**B**), IFN-γ (**C**), IL-17 (**D**), and IL-10 (**E**). (**F**) Overall visualization of groups receiving prime or prime/boost with rBCG PspA-PdT/rPspA-PdT regarding cytokine production in the lungs. All results are represented by the means± SD. * *p* values < 0.05 were considered statistically significant.

**Figure 4 vaccines-12-00122-f004:**
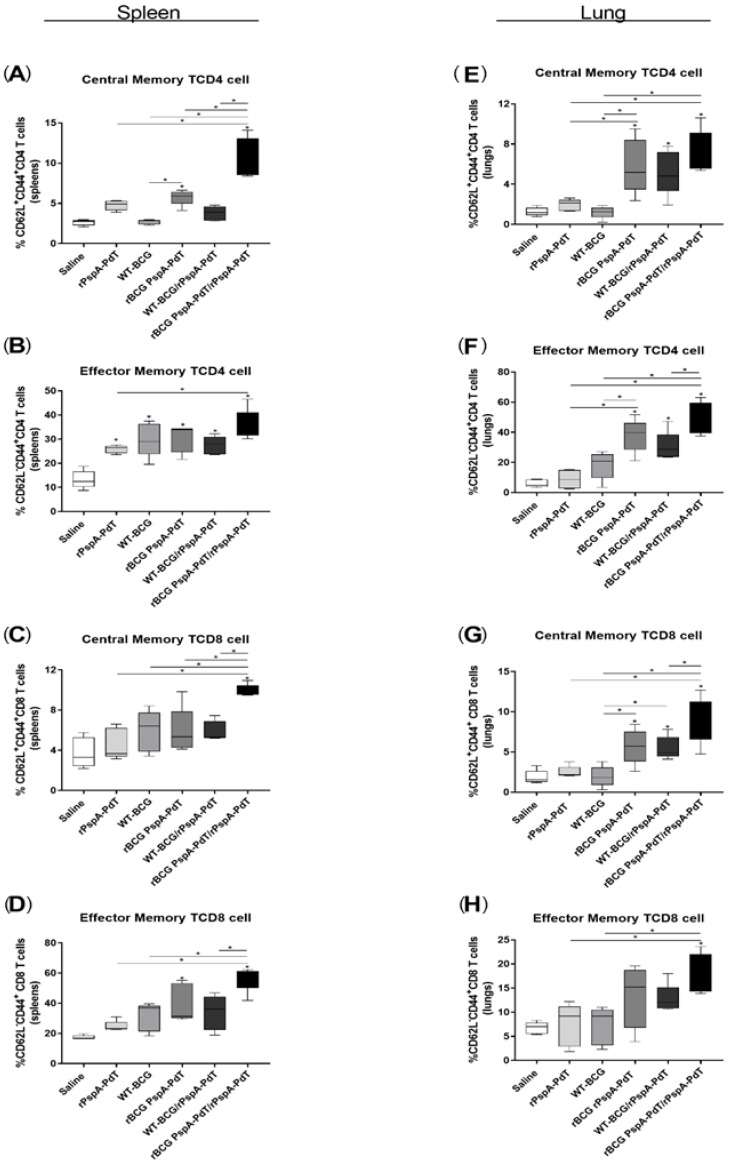
rBCG PspA-PdT/PspA-PdT promotes the development of Memory and Effector T cells in the spleen and lungs in the neonate mice. Neonate mice were vaccinated with WT-BCG, rBCG PspA-PdT, or rPspA-PdT. After 7 days, mice received a booster dose with the rPspA-PdT. After 9 days of the booster, subsets for Memory T cells in the spleens and lungs were evaluated. Central Memory T cells were characterized as CD4^+^CD62L^+^CD44^+^ in the spleens (**A**) and lungs (**E**) or CD8^+^CD62L^+^CD44^+^, respectively (**C**,**G**). Effector Memory T cells were characterized as CD4^+^CD62L-CD44^+^ in the spleens (**B**) and lungs (**F**) or CD8^+^CD62L^+^CD44^+^, respectively (**D**,**H**). * *p* values < 0.05 were considered statistically significant. Results are represented by the means ± SD.

**Figure 5 vaccines-12-00122-f005:**
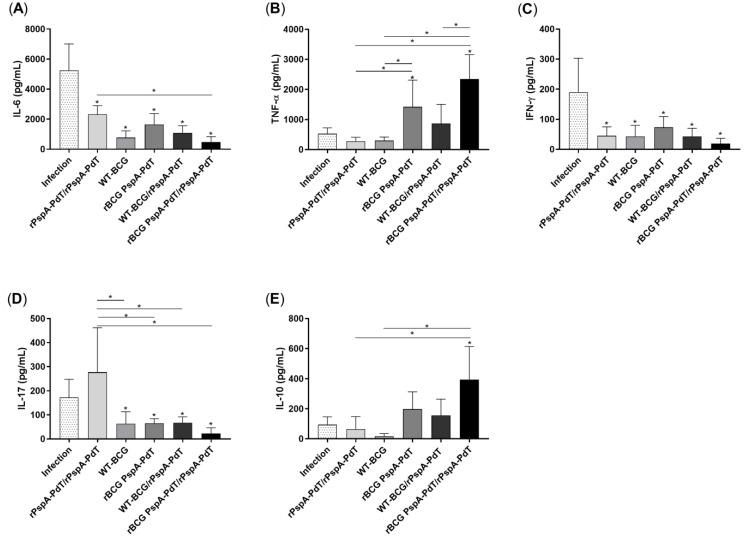
rBCG PspA-PdT/PspA-PdT reduces cytokines in the lungs of pneumococcal-infected mice. Neonate mice were vaccinated with WT-BCG or rBCG PspA-PdT or rPspA-PdT. After 7 days, mice received a booster dose with the rPspA-PdT. Nine days after the booster dose, the mice were exposed to the WU2 pneumococcal strain via the intranasal route. Three days after the challenge, cytokine levels were quantified in the supernatant of BALF via mouse Th1/Th2/Th17 cytometric bead array (CBA). (**A**) IL-6, (**B**) TNF-α, (**C**) IFN-γ, (**D**) IL-17, and (**E**) IL-10. All results are represented by the means ± SD. * *p* values < 0.05 were considered statistically significant.

**Figure 6 vaccines-12-00122-f006:**
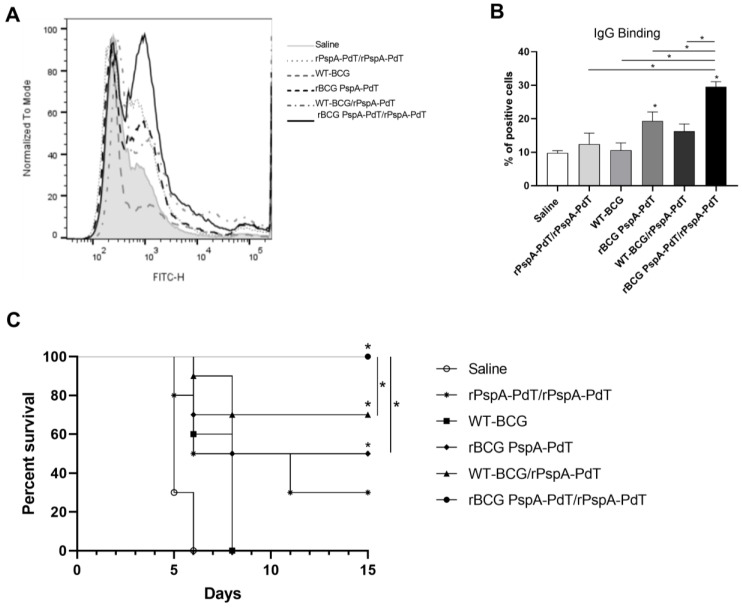
Prime/boost strategy using rBCG PspA-PdT/rPspA-PdT provides protection against a lethal pneumococcal challenge. Neonate mice (5 days old) were vaccinated with WT-BCG, rBCG PspA-PdT, or rPspA-PdT. After 7 days, mice (12 days old) received a booster dose with the rPspA-PdT. Nine days later, mice (21 days old) were intranasally challenged with a virulent pneumococcal strain (WU2, 10^6^ CFU). (**A**,**B**) Three days after the challenge, BALF was collected. Pneumococcal WU2 strain was cultured with BALF samples obtained from mice that were both immunized and infected. This is followed by incubation with FITC-conjugated antibody against mouse IgG. The percentage of positive cells (FITC-IgG) was analyzed via flow cytometry. (**C**) Survival of animals was monitored for 15 days. * *p* values < 0.05 were considered statistically significant.

## Data Availability

The data presented in this study are available on request from the corresponding author.

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
