# Peer review of "Robust Immune Response and Protection against Lethal Pneumococcal Challenge with a Recombinant BCG-PspA-PdT Prime/Boost Scheme Administered to Neonatal Mice"

_vaccines, 2024, doi:10.3390/vaccines12020122_

Round 1

Reviewer 1 Report

Comments and Suggestions for Authors

Overall Comments:

Pneumococcal infections are a significant global health problem, particularly in developing countries, where vaccination coverage is limited and serotype replacement and antibiotic resistance are increasing. Conjugate pneumococcal vaccines have limitations in protecting neonates and providing coverage against all serotypes. Protein-based pneumococcal vaccines have been studied as alternative options. Pneumococcal surface protein A (PspA) and pneumolysin (Ply) are two well-investigated proteins with high immunogenicity and protective efficacy. The authors previously developed a recombinant BCG expressing a fusion protein of PspA and a genetically detoxified form of Ply (PdT), which showed promising results in adult animals. In this study, the authors aimed to investigate the protective effect of this prime/boost strategy in neonate mice.

Overall, the manuscript is well written, the methods described in good detail, and the figures with corresponding legends provide the data in a clear form. The conclusions of the study are supported by the data presented, and are clearly stated. The reviewer believes that this manuscript with interesting information for readers and researchers involved in the pneumococcal infection.

To strengthen the manuscript, I suggest the following revisions:

1)         The background introduction in the abstract is too brief and lacks detail. You need to provide a more comprehensive overview of the topic to engage readers.

2)         Provide more context and background information on the topic to make the article more accessible to a wider audience.

3)         Address the limitations of the study, such as the reliance on biased datasets and the absence of a proper sampling design. Discuss potential implications of these limitations on the interpretation of the findings.

Minor comments:

   Figure 2, where is the subtitle "D" in the lung panel?

Comments on the Quality of English Language

Moderate editing of English language required.

Author Response

Reviewer 1

We appreciate the positive appreciation of the manuscript and thank the suggestions, which are answered bellow:

Comment 1- The background introduction in the abstract is too brief and lacks detail. You need to provide a more comprehensive overview of the topic to engage readers.

Response: Thank you for pointing this out. We agree with this comment, and we have, accordingly, revised the abstract to emphasize this point.  Lines 10-13. 

Comment 2:  Provide more context and background information on the topic to make the article more accessible to a wider audience.

Response: Thank you for pointing this out. We have expanded the introduction to provide more context. Lines 38-41; 50-55; 66-73.

Comment 3:  Address the limitations of the study, such as the reliance on biased datasets and the absence of a proper sampling design. Discuss potential implications of these limitations on the interpretation of the findings.

Response: Thank you for pointing this out. One of the limitations of this study is the reduced number of studies with neonates in the literature, especially in the BALF, which provided few options for proper sampling design. We actually adopted alternative designs, more appropriate for our objectives, which reduces the possibility of comparison with previous data. Another important factor was the fact that obtaining a reasonable amount of neonate mice at the same time requires a very careful coordination for their birth to be able to obtain several litters at the same time. Furthermore, the amount of blood and BALF obtained is much more limited than in adult mice, restricting the number of samples available for the analysis. These limitations were included in the discussion (line 434-443). Another limitation is the variability of PspA. We included a discussion and possible solutions to this limitation in the discussion, Lines 443-455.

Minor comment: Figure 2, where is the subtitle "D" in the lung panel?

Response: We thank you for pointing this out. We have included the D in the Figure.

Reviewer 2 Report

Comments and Suggestions for Authors

The current article by Tretini et al. examines neonate immunization efficacy using different strategies. The authors focus on the efficacy of using BCG in combination with pneumococcal proteins to prime neonate mice during initial immunization followed by a boost of recombinant pneumococcal proteins. Authors examine antibody titers, antibody binding to whole pneumococcal cells, production of memory cells and cytokines, along with protective effects of their immunization strategy. Overall the article is well written and is an expansion on previous work done in adult mice. The experiments performed and analysis are presented well but there are a few issues concerning the presentation that should be addressed and are detailed below.

Major:

More detail regarding the actual immunogen should be included. BCG is a live attenuated vaccine but it is stated that a recombinant BCG was created. It is unclear what is actually being used for the immunization. Is it BCG that has been modified to express the PspA-PdT fusion on the surface of the attenuated BCG strain? Is it a specific immunogen from BCG fused to PspA-PdT or is it BCG given in conjunction with purified PspA-PdT? This needs to be made clear. In addition to this what PspA family is being used in the rPspA antigen? PspA is highly variable and cross protection against other families is not common. Is the rPspA from WU2 which is used in infection studies?

In method section 2.5 for antibody binding more detail is needed. Is WU2 incubated in undiluted sera? Is the FITC-conjugated antibody anti-mouse IgG(H+L) since there are multiple isotypes being examined? How long was the incubation for the sera and secondary.

Minor:

Ln. 44 “(Ply) is an extremely…”

Ln. 85 change star to state

Ln 148 remove “bacteria”

Figure 1C. The scale is cut off slightly

Ln 263. In the figure legend is 106 correct or was 10^6 meant?

Ln 301-302 “balances as a possible protective.” This statement is not clear. If trying to say that the previously observed IgG1/IgG2c balance may not be as protective against infection as the IgG2c shift seen in the current study then this is not made clear.

Ln 322. Italicize “L. casei”

Ln 322. Alpha symbol is incorrect for TNF-alpha

Ln 323. Italicize “S. pneumoniae”

Comments on the Quality of English Language

Included in comments and suggestions

Author Response

Comment 1: More detail regarding the actual immunogen should be included. BCG is a live attenuated vaccine but it is stated that a recombinant BCG was created. It is unclear what is actually being used for the immunization. Is it BCG that has been modified to express the PspA-PdT fusion on the surface of the attenuated BCG strain? Is it a specific immunogen from BCG fused to PspA-PdT or is it BCG given in conjunction with purified PspA-PdT? This needs to be made clear. In addition to this what PspA family is being used in the rPspA antigen? PspA is highly variable and cross protection against other families is not common. Is the rPspA from WU2 which is used in infection studies?

Response: Thank you for pointing this out, this information is actually very important. We developed a recombinant BCG (rBCG) expressing the rPspA-PdT fusion protein (rBCG PspA-PdT). The PspA used in the study was a PspA clade 2, family 1, as described in reference 27. This information was inserted in the manuscript, in section Methods 2.1. (Lines 94-98). Furthermore, we included a paragraph on the variability of PspA and how we intend to deal with this, Introduction, Lines 50-55.

 Comment 2:  In method section 2.5 for antibody binding more detail is needed. Is WU2 incubated in undiluted sera? Is the FITC-conjugated antibody anti-mouse IgG(H+L) since there are multiple isotypes being examined? How long was the incubation for the sera and secondary?

Response 2: Thank you for pointing this out. This is a limitation in this study. Due to the limited amount collected of both serum and BAL from mice, we did not perform the binding assay with IgG isotypes (IgG1 and IgG2c) and IgA. Serum and BAL samples were not diluted, and FITC-conjugated antibody anti-mouse IgG was incubated for 30 min in the dark. These statements were inserted in Method section 2.5 (Lines 123 and 125) of the manuscript for a better understanding of the assay.

 Minors Comments:  

Response 3: We agree with these suggestions. We have made all the changes suggested in the manuscript:

Ln. 44 “(Ply) is an extremely…” We have modified this section, Lines 58-60

Ln. 85 change star to state, This is corrected in Line 112

Ln 148 remove “bacteria”, This is corrected in Line 202.

Figure 1C. The scale is cut off slightly, This was corrected.

Ln 263. In the figure legend is 106 correct or was 10^6 meant? This was corrected

Ln 301-302 “balances as a possible protective.” This statement is not clear. If trying to say that the previously observed IgG1/IgG2c balance may not be as protective against infection as the IgG2c shift seen in the current study then this is not made clear.

We have modified the statements to clarify, Lines 167-173.

Ln 322. Italicize “L. casei” . This is corrected in Line 412

Ln 322. Alpha symbol is incorrect for TNF-alpha. This is corrected in Line 412.

Ln 323. Italicize “S. pneumoniae”. This is corrected in Line 413.

Reviewer 3 Report

Comments and Suggestions for Authors

This is an interesting study examining the protective effect of a novel recombinant vaccine against pneumococcal infection. Several comments are made for the authors consideration:

1. The title sounds more like a sentence than a title. I suggest the authors rephrase the title to: Robust Immune Response and Protection against Lethal Pneumococcal Challenge with a Recombinant BCG-PspA-PdT Prime/Boost Scheme administered to Neonatal Mice.

2. The abstract is good but lacks some important details, such as (1) the number and age of included mice, (2) was there a control group?, (3) what is lethal pneumococcal infection? In other words, what was pneumococcal strain used?

3. Introduction (lines 63-67): This paragraph summarizes the findings of the current study as if it's the conclusion. The last paragraph of the introduction should only clearly indicated the objectives/aim of the study without stating the results, which will be shown later in the paper.

4. 2.1 Pneumococcal strain: Please add a brief description or a background supported by evidence from the literature if available on why this specific strain is considered lethal.

5. Line 82: Change "Mice previously immunized" to "All mice groups" since non-immunized negative controls have also received the infection.

6. Line 133: "*p < 0.05 versus control or indicated groups" this sentence seems to be copied from a footnote of a figure. It should be rewritten to: A p < 0.05 indicated statistical significance.

7. All Figures: The footnote says "*p values ≤ 0.05 were considered statistically significant." However, on line 133 (as in the comment above), it was a p < 0.05 and not ≤ 0.05. Please be consistent. The correct statistically significant p value should be only less than 0.05 and not equal to it.

8. Discussion: Please add some details regarding the pneumococcal strains that this recombinant vaccine can protect from. Details on the geographic distribution of the strain used in this study would also be appreciated.

9. Discussion: Do you have any idea for how long can this recombinant vaccine remain effective when administered to a human neonate? How many years do the antibodies remain circulating? If not, then it could be worth mentioning as a suggested aspect to be explored in future human studies.

Author Response

Reviewer 3 

Comments 1: The title sounds more like a sentence than a title. I suggest the authors rephrase the title to: Robust Immune Response and Protection against Lethal Pneumococcal Challenge with a Recombinant BCG-PspA-PdT Prime/Boost Scheme administered to Neonatal Mice.

Response 1: Thank you for the sugestion. The title was changed.

Robust Immune Response and Protection against Lethal Pneumococcal Challenge with a Recombinant BCG-PspA-PdT Prime/Boost Scheme administered to Neonatal Mice.”

Comments 2:  The abstract is good but lacks some important details, such as (1) the number and age of included mice, (2) was there a control group?, (3) what is lethal pneumococcal infection? In other words, what was pneumococcal strain used?

Response: We agree that this information was missing and we have modified the Abstract accordingly, Lines 16-22. More detailed information was included in the Methods section 2.2., 100-107.  

Comments 3: Introduction (lines 63-67): This paragraph summarizes the findings of the current study as if it's the conclusion. The last paragraph of the introduction should only clearly indicate the objectives/aim of the study without stating the results, which will be shown later in the paper.

Response: The conclusion was removed from the Introduction, and only the objective was written in the paragraph.

Introduction, Line 84.

Comments 4: 2.1 Pneumococcal strain: Please add a brief description or a background supported by evidence from the literature if available on why this specific strain is considered lethal.

Response: This information was added to the Methods section, 2.1., Lines 88-93..

Comments 5: Line 82: Change "Mice previously immunized" to "All mice groups" since non-immunized negative controls have also received the infection.

Response 5: Thank you for suggestion. We changed the sentence, Line 109.

Comments 6: Line 133: "*p < 0.05 versus control or indicated groups" this sentence seems to be copied from a footnote of a figure. It should be rewritten to: A p < 0.05 indicated statistical significance

Response: Thank you for the suggestion. We modified this in the methods section, Line 160.

Comments 7: All Figures: The footnote says "*p values ≤ 0.05 were considered statistically significant." However, on line 133 (as in the comment above), it was a p < 0.05 and not ≤ 0.05. Please be consistent. The correct statistically significant p value should be only less than 0.05 and not equal to it.

Response: Thank you for the suggestion. We modified all footnotes of figures to: “p < 0.05 “

Comments 8: Discussion: Please add some details regarding the pneumococcal strains that this recombinant vaccine can protect from. Details on the geographic distribution of the strain used in this study would also be appreciated.

Response: We included this information in Methods, section 2.1., Lines 90-93. We also discussed this issue in the Introduction, Lines 50-55, and in the Discussion, Lines 444-455.

Comments 9: Discussion: Do you have any idea for how long can this recombinant vaccine remain effective when administered to a human neonate? How many years do the antibodies remain circulating? If not, then it could be worth mentioning as a suggested aspect to be explored in future human studies.

Response: Thank you for raising this important point. We included a discussion on this point inn the Conclusion, Lines 463-466.

Reviewer 4 Report

Comments and Suggestions for Authors

The manuscript entitled” Neonate mice immunized with a recombinant BCG-PspA-PdT prime/boost scheme displays a robust immune response and protection against lethal pneumococcal challenge” is an interesting piece of work that demonstrates the applications of recombinant vaccines towards pneumonia. However, the quality of the manuscript may be strengthened by addressing the following queries:

1.      In the abstract section, authors should provide the degree of improvement in terms of data value to showcase the findings of the present studies.  

2.      In the introduction section, line 44, authors should discuss the biochemical description of pneumo-lysin protein.  

3.      In figure 1c, the content on the x-axis is not properly presented. Authors should correct the presentation of the data.

4.      Figure 2, the trends of results are interesting and show a similar pattern irrespective of concentration. Authors should discuss one statement why the spleen is giving the better antibodies response.

5.      The back reference section is not uniform. Authors should adhere to the author guidelines of this journal of repute for uniformity in the reference citation.

Author Response

Reviewer 4 

Comments 1: In the abstract section, authors should provide the degree of improvement in terms of data value to showcase the findings of the present studies.   

Response: Thank you for pointing this out. The data was included, Line 20-22.

Comments 2:  In the introduction section, line 44, authors should discuss the biochemical description of pneumo-lysin protein. 

Response: We have included more information on the Pneumolysin protein, Line 58-60.

Comments 3: In figure 1c, the content on the x-axis is not properly presented. Authors should correct the presentation of the data.  

Response: We thank you for pointing this out. We corrected the Figure 1C axis.    

Comments 4:  Figure 2, the trends of results are interesting and show a similar pattern irrespective of concentration. Authors should discuss one statement why the spleen is giving the better antibodies response.

Response: This difference in cellular concentration between the organs studied in Figure 2 (spleen and lung) is related to the fact that the spleen is a hemopoietic organ (secondary organ), with the function of maturation and storage of immune system cells. This information was included in Lines 216-218.

Comments 5:  The back reference section is not uniform. Authors should adhere to the author guidelines of this journal of repute for uniformity in the reference citation.

Response: We thank the reviewer for pointing this out. We revised have revised the whole Reference section.

Round 2

Reviewer 3 Report

Comments and Suggestions for Authors

Thank you to the authors for incorporating the suggested edits. I have no further comments.